# Coinfections and Superinfections Associated with COVID-19 in Colombia: A Narrative Review

**DOI:** 10.3390/medicina59071336

**Published:** 2023-07-20

**Authors:** Diana Dueñas, Jorge Daza, Yamil Liscano

**Affiliations:** 1Grupo de Investigación en Salud Integral (GISI), Departamento Facultad de Salud, Universidad Santiago de Cali, Cali 760035, Colombia; diana.duenas00@usc.edu.co; 2Grupo de Investigación de Salud y Movimiento, Programa de Fisioterapia, Facultad de Salud, Universidad Santiago de Cali, Cali 760035, Colombia; jorge.daza01@usc.edu.co

**Keywords:** COVID-19, SARS-CoV-2, coinfections, antimicrobials, Colombia, epidemiology, coinfections, superinfections

## Abstract

The COVID-19 pandemic has had significant impacts on healthcare systems around the world, including in Latin America. In Colombia, there have been over 23,000 confirmed cases and 100 deaths since 2022, with the highest number of cases occurring in females and the highest number of deaths in males. The elderly and those with comorbidities, such as arterial hypertension, diabetes mellitus, and respiratory diseases, have been particularly affected. Coinfections with other microorganisms, including dengue virus, *Klebsiella pneumoniae*, and *Mycobacterium tuberculosis*, have also been a significant factor in increasing morbidity and mortality rates in COVID-19 patients. It is important for surveillance systems to be improved and protocols to be established for the early detection and management of coinfections in COVID-19. In addition to traditional treatments, alternatives such as zinc supplementation and nanomedicine may have potential in the fight against COVID-19. It is also crucial to consider the social, labor, educational, psychological, and emotional costs of the pandemic and to address issues such as poverty and limited access to potable water in order to better prepare for future pandemics.

## 1. Introduction

In December 2019, in Wuhan, China, an unidentified coronavirus emerged, causing a major outbreak in many cities and rapidly spreading globally. This new coronavirus, known as severe acute respiratory syndrome coronavirus (SARS-CoV) 2 (SARS-CoV-2), is a virus containing a genome with 29,903 nucleotides and 29 proteins, belonging to the family Coronaviridae, subfamily Coronavirinae, and is the main cause of severe acute respiratory syndrome, also known as COVID-19 [1,2,3]. By the end of 2021 and early 2022, COVID-19 had infected 224 million people, and 4.6 million had died globally [4].

In addition, microbial coinfections and superinfections [5] had occurred, influenced by factors such as the potentiation of pathogenesis and the increased risk of morbidity and mortality of patients with COVID-19. Coinfection can be defined as the recovery of other pathogens in a patient with an infection within 48 h of admission [6,7], and superinfection occurs when a patient has clinical signs and symptoms of pneumonia or bacteremia combined with a positive culture of a new pathogen from a lower respiratory tract or blood specimen obtained ≥48 h after admission [8]. In other words, coinfection occurs simultaneously with the spread of the microorganism, while superinfection develops after the initial infection [9]. 

Latin America had been one of the most severely affected regions by the COVID-19 pandemic, accounting for 25% of global infections. Moreover, of the ten countries with the highest mortality rates worldwide, eight were from this region, including Colombia [10]. In a study conducted in Colombia, a high incidence of early mortality associated with COVID-19 24 h after hospital admission was reported [11]. However, studies related to the incidence of coinfection and superinfection in patients with COVID-19 are generally limited, especially in Latin American countries [5,12,13]. It was observed that severely ill COVID-19 patients, especially those in the intensive care unit (ICU), were more prone to secondary infections, owing to the increased use of prophylactic or therapeutic antibiotics whose task is to ensure the successful eradication of susceptible pathogens [14]. However, antibiotic misuse poses a threat due to the increase in the number of antibiotic-resistant microorganisms; additionally, it has a negative impact on the host microbiota. Clinical evidence suggests that inappropriate empirical use of antimicrobials may be associated with increased morbidity and mortality [9,15,16]. Therefore, the objective of this review is to describe scientific evidence on SARS-CoV-2-related coinfections and superinfections and their relevance in patients with COVID-19 in Colombia.

## 2. Methodology

A comprehensive literature search was conducted using databases such as PubMed, Scopus, SciELO, and Web of Science. The search was based on keywords related to COVID-19, SARS-CoV-2, coinfections, superinfections, Colombia, and COVID-19 treatments ((SARS-CoV-2 OR COVID-19) AND Colombia AND coinfection AND superinfection AND therapy). Relevant data corresponding to each section of the manuscript, such as definitions, pathophysiology, and treatment of SARS-CoV-2, were extracted. The extracted data were then analyzed to identify common themes, patterns, and trends related to coinfections and superinfections. The findings were organized in a narrative format, highlighting key points, similarities, and differences among the studies.

Definitions included the following:

Coinfections: “Coinfections with other microorganisms such as bacteria, fungi, and other viruses are commonly associated with respiratory viral infections. Coinfections are directly linked to increased rates of morbidity and mortality, thus requiring early diagnosis and specific treatment” [17].

Superinfection: “Superinfection is diagnosed when patients present with clinical signs and symptoms of pneumonia or bacteremia combined with a positive culture of a new pathogen from a lower respiratory tract sample (including sputum, transtracheal aspirates, or bronchoalveolar lavage fluid) or blood samples taken ≥48 h after admission” [8].

## 3. SARS-CoV-2

Coronaviruses are approximately 80–220-nm-diameter enveloped viruses [2]. The viral genome encodes five structural proteins which are encoded within the 3’ end, namely, spike protein (S), envelope (E), membrane (M), nucleocapsid (N), and hemagglutinin esterase (HE) [1,2,3,18,19]. Protein S is a transmembrane glycoprotein that facilitates viral envelope binding to angiotensin-converting enzyme 2 (ACE-2) receptors expressed on the surface of host cells; it also forms protruding homotrimers on the viral surface, and this protein comprises two functional subunits: receptor binding (S1) and cell membrane fusion (S2) [1,2,18]. The E protein is the smallest protein in the SARS-CoV-2 structure, and its function is not necessary for replication. However, it plays a huge role in pathogenesis since it helps in the assembly and liberation of virions [1,18]. The M protein is the most abundant protein in the virion, and it was suggested that this protein plays a role in RNA packaging and promoting the assembly and budding of viral particles through interaction with N and accessory proteins 3a and 7a [1,2,18]. The N protein packages genetic material and binds to the viral genome in a bead-on-a-string conformation. Consequently, it modifies host-cell RNA processing, alters the TGF-β pathway by blocking apoptosis, and promotes binding of the transcription factor NF-κβ to the COX-2 promoter, leading to an inflammatory response. Notably, it is also involved in RNA replication and immune evasion [2,18]. The HE protein acts as a hemagglutinin, binds to sialic acids on surface glycoproteins, contains acetyl esterase activity, and helps the virus spread through the mucosa [1]. Among these five proteins, the most important ones are protein N and protein S. While the former helps in the development of the capsid, the latter allows the virus to bind to target cells. Meanwhile, the most complex component is the receptor-binding domain (RBD) in the S protein because six RBD amino acids are needed to bind to the ACE-2 receptor and harbor SARS-CoV-2-like coronaviruses [1,2]. 

The RBDs of SARS-CoV-2 have a stronger attraction or affinity for the ACE-2 receptor compared to the RBDs of SARS-CoV. Furthermore, in the case of SARS-CoV-2, a significant portion of the RBDs is in the bound state, meaning they are attached or bound to the ACE-2 receptor. This binding state leads to a comparable or potentially lower affinity for the receptor when compared to SARS-CoV [3]. 

SARS-CoV-2 entry into host cells and the release of its genomes depend on a sequence of steps, but it should be noted that four of the structural proteins it possesses (S, N, M, and E) allow it to gain access to target cells. For entry, the virus requires binding of the S protein to ACE-2 on the cell’s plasma membrane. However, this protein must be cleaved by the transmembrane serine protease 2 (TMPRSS2) of the cell membrane into the two subunits—S1 containing the RBD to ACE-2 and S2 facilitating viral fusion—although this cleavage can also occur by cathepsin-L in the endosomes, which helps to infect cells without TMPRSS2, but this is a slower process [18,20,21]. The literature suggests that the modified RBD residues of protein S in SARS-CoV-2 contribute to its high pathogenicity and transmissibility compared to SARS-CoV. Moreover, the presence of the polybasic furin cleavage site is not observed in other coronaviruses. Thus, this facilitates efficient cleavage of the S protein by furin and other proteases. Even the “S trimer” exists in a partially open state in highly pathogenic coronaviruses [20]. After the virus enters the target cell cytoplasm, uncoated genomic RNA is translated into polyproteins (pp1a and pp1ab), which are then assembled into replicating or virus-induced double-membrane vesicle transcription complexes. Subsequently, these complexes replicate and synthesize a nested set of genomic RNA by genome transcription, which encodes structural proteins (M, E, and N) and some accessory proteins. The endoplasmic reticulum and the Golgi complex mediate the binding of the newly formed viral particles, then finally, the new virions leave the cell by exocytosis [18].

## 4. SARS-CoV-2 Immunology

A special feature of SARS-CoV-2 is the inhibition of receptor signaling pathways responsible for triggering antiviral immunity, mainly pattern recognition receptors (PRRs) tasked to recognize molecular patterns associated with pathogens or cellular damage (PAMPs and DAMPs). Figure 1 shows the immunology against SARS-CoV-2. PAMPs are associated with microbial pathogens [22]. The main PRRs for viral recognition are toll-like receptors 3 (TLR3) and 7 (TLR7) in the endosome or the cytosolic sensors retinoic acid-inducible gene 1 (RIG-I) and melanoma differentiation-associated gene 5 (MDA5). Activation of these PRRs is mainly associated with IFN production [22,23]. RGI-I and MDA5 activate the mitochondrial antiviral signaling protein of the downstream adaptor in the mitochondria, followed by activation of TNF receptor-associated protein [23]. SARS-CoV-2 has the ability to suppress the production and function of type 1 IFNs, triggering IFN-stimulated genes [22,23].

Similarly, in response to coronavirus infection, humans produce TCD4+ lymphocytes, TCD8+ lymphocytes, and specific antibodies, which have protective functions against viral infections; however, these functions and their importance vary according to the viral infection. TCD4+ cell responses to primary *SARS-CoV-2* infection are more prominent than those of TCD8+, whereas the presence of specific TCD8+ cells has been associated with better outcomes in COVID-19 [24]. Humoral response to SARS-CoV-2 involves neutralizing antibodies (nAb) specific to viral epitopes. N protein epitopes are conserved among different coronaviruses, prompting the generation of cross-reactive antibodies. However, nAb targets protein S and its RBD region to neutralize the coronavirus. In turn, they protect against future infections [3,24]. Moreover, SARS-CoV-2 infection can lead to a reduction in lymphocytes [23].

It is worth mentioning that the virus has an impact on the mechanisms of cellular stress activation in immunocompetent cells because it causes the activation and apoptosis of lymphocytes and macrophages as well as immunosuppression [22]. The cytokine storm correctly reflects the immune response in patients with COVID-19. The observed elevated IL-6 levels are considerably low. However, there are dynamic changes in the concentration of many cytokines, including IL-6. In addition, kynurenines, molecules related with immunosuppression, are elevated in severe COVID-19 [23]. 

## 5. Impact of SARS-CoV-2in Colombia

Nearly two decades after the emergence of SARS-CoV, SARS-CoV-2spread rapidly after the first reported case in December 2019 and became a serious global health crisis [25]. Noticeably, the most affected countries were industrialized ones with strong public health systems and advanced medical facilities that have been severely strained in the course of the pandemic [26]. However, Latin America’s health care systems have been significantly disrupted as SARS-CoV-2 spread around the world [27]. These nations opted for strict quarantine and the promotion of self-care, following WHO guidelines, whose objective was to separate potential carriers of the virus from uninfected individuals and thus reduce the spread of the virus, corresponding to epidemiological fences, but the social, labor, educational, psychological, and emotional costs were high [28]. 

In particular, the pandemic adversely affected the elderly and the disadvantaged the most. While all age groups were susceptible to SARS-CoV-2 infection, older adults suffered a higher risk of mortality [29]. Latin American countries faced an extremely protracted challenge, as factors such as poverty and limited access to potable water, among others, continued to be a critical point in the development of the region. In addition, there was a fundamental lack of preparedness to deal with a pandemic [29]. 

In early March 2020, the first positive cases in Colombia were reported in the city of Bogota, causing great concern among the people and leading to the implementation of weak measures to mitigate transmission. In turn, the government had to declare mandatory containment on 25 March 2020, which remained in force, totally or partially, until the beginning of 2022 [30,31,32]. For some authors, the health and security system, the economic sector, the social context, and people’s mental health were the most critically impacted by the pandemic in the country, with the health and social security system being most affected due to the rapid chains of transmission [30]. 

In Colombia, from 3 January 2020 to 5 July 2023 at 4:01 p.m. CEST, there have been 6,373,599 confirmed cases of COVID-19 with 142,836 deaths reported to the WHO. As of 2 June 2023, a total of 90,506,612 vaccine doses have been administered [33].

Colombia had a case fatality rate of 2.5% with an accumulated positivity rate of 22.6%, with the highest number of cases occurring in females (53%) and the highest number of deaths in males (60.7%). Furthermore, the highest frequency of deaths was found among those between 40 and 90 years old with 389,736 deaths, peaking from 70 to 79 years old with 34,924 deaths. Among the comorbidities affecting the deceased, the most frequent were arterial hypertension (6416 deaths), diabetes mellitus (3901 deaths), respiratory diseases (2421 deaths), renal diseases (2226 deaths), obesity (1910 deaths), cardiac diseases (2083 deaths), and cancer (1115 deaths) [34].

Colombia faced six “waves,” and their delimitations correlated with the restriction and relaxation of measures such as social distancing [35]. It should be noted that, according to clinical evidence, most of the cases reported in the second, third, and fourth waves were mild to moderate with lower hospital admission requirements and short hospital stays, i.e., there was a higher survival rate from COVID-19, but findings showed a large increase in the number of confirmed cases in those periods. According to the Instituto Nacional de Salud (INS), a total of 5,823,994 symptomatic cases, 488,614 asymptomatic cases, and 141828 deaths have been recorded in the six waves that have occurred in the country. The fourth and fifth waves reported higher numbers of symptomatic COVID-19 cases, with 3 January 2022 (belonging to the fifth wave) being the day with the highest number of confirmed cases (41,474). The second and fourth waves had higher numbers of asymptomatic cases reported in consultation, with 15 June 2021 (belonging to the fourth wave) being the day with the highest number of registered asymptomatic cases (5315). The fourth wave saw the highest number of COVID-19 deaths reported [34]. 

In addition, the INS reports 22,968 complete genomes sequenced with 267 lineages identified from 25,066 tests. Among the lineages found in the first sampling on 5 September 2021, there was a similar prevalence between the Mu (55%) and Delta (45%) variants. Patiño et al. [36] analyzed the effective reproduction number (Rt) of *SARS-CoV-2*, the virus that causes COVID-19, in Cali, Colombia between April and July 2021. The study found that Rt values were higher during the period of frequent protests compared to the preceding and following months. Genomic analysis revealed the circulation of 16 different lineages of SARS-CoV-2, including variants of concern (VOCs) and variants of interest (VOIs). The study suggests that the spread of highly virulent strains of SARS-CoV-2 in Cali and other parts of Colombia was facilitated by the limited biosecurity strategies during the period of political turmoil and social demonstrations, as well as the movement of large numbers of people into and out of the city. The Mu variant is thought to have been introduced to Cali on two separate occasions, and may have contributed to the 55% increase in the number of reported cases during the protests. 

By 2 January 2022, 62% belonged to the delta variant and 38% to the Omicron variant, which later became dominant with 100% in circulation on 13 March 2022. On 8 May 2022, two sublineages of the Omicron strain, BA.2.12.1 (20%) and BA.2.x (80%), were presented. Finally, on the last sampling date of 7 August 2022, two sublineages of the Omicron variant, BA.4 (40%) and BA.5 (60%), were revealed [34]. Finally, until 18 December 2022, the omicron BQ.1.x subvariant was reported with 90% prevalence in Colombia and the XBB variant with 10%, the latter being the most transmissible variant [37]. 

## 6. Coinfections Associated with SARS-CoV-2 in Colombia

Coinfections with microbial pathogens have played an important role in increasing the morbidity and mortality rate in pandemics, and SARS-CoV-2 is no exception [38]. Coinfection between other microorganisms and this new coronavirus is an important factor to take into account in COVID-19, as it may increase the difficulties in diagnosis, management, and prognosis, and even worsen symptoms and mortality [12]. It should be noted that a wide spectrum of atypical presentations were found to be associated with COVID-19, which complicated critically ill immunocompromised patients [39]. Table 1 lists the most frequent coinfections; however, bacterial and viral coinfections were most frequently reported in patients with COVID-19 and had proportions as high as 50% among non-survivors. However, severe cases of COVID-19 related to fungal infections were documented, especially representing a major threat to life in ICU patients [16,38,40,41]. Mechanical ventilators and catheters are risk factors for nosocomial infections [14,40].

The source and specific nature of these infections have not yet been fully explored, but there is evidence to suggest that multidrug-resistant bacteria, such as *Staphylococcus aureus*, *Streptococcus pneumoniae*, *Klebsiella pneumoniae*, and *Escherichia coli*, are among the pathogens believed to be causative agents. *Acinetobacter* spp., *Enterobacter* spp., *Enterococcus* spp., and *Pseudomonas* spp. are also associated with hospital-acquired infections. Similarly, several studies have indicated that coinfection with fungi such as *Aspergillus* spp. and *Candida* spp. can increase mortality rates [14,41,42]. In addition, coinfections with SARS-CoV-2 and other respiratory viruses are unusual. They have been observed from 3.2% to 22.4% of the time, with rhinovirus–enterovirus (6.9%) and respiratory syncytial virus (5.2%) being the most commonly reported [43].

**Table 1 medicina-59-01336-t001:** Coinfections with *SARS-CoV-2* worldwide. HIV: Human Immunodeficiency Virus; MRSA: Methicillin-Resistant Staphylococcus Aureus.

Author	Coinfection	Country
García-Vidal et al., 2021 [5]	Influenza A Influenza B *Virus sincitial respiratorio* *Streptococcus pneumoniae* *Staphylococcus aureus* *Haemophilus influenzae* *Moraxella catarrhalis*	Spain
Acosta-Pérez et al., 2022 [13]	Dengue	Colombia
Motta et al., 2022 [43]	Adenovirus	Colombia
Forero-Peña et al., 2022 [44]	Malaria	Venezuela
Jeong et al., 2022 [45]	*Acinetobacter baumannii* *Pseudomonas aeruginosa* *Staphylococcus aureus* *Klebsiella pneumoniae* *Escherichia coli* *Bocavirus* *Influenza B*	South Korea
Palou et al., 2021 [46]	Rhizopus spp.	Honduras
Schulte et al., 2021 [47]	Dengue	Brazil
Almeida et al., 2021 [48]	*Candida auris*	
Santana et al., 2020 [49]	*Aspergillus penicillioides*	
Torres-Serrano et al., 2021 [50]	*Cryptococcus neoformans*	Colombia
Ortiz-Martínez et al., 2021 [51]	*Mycobacterium tuberculosis*	
Martínez-Montalvo et al., 2021 [52]	*Pneumocystis jirovecii.*	
Álzate- Ángel et al., 2020 [53]	VIH	
Mejía- Parra et al., 2021 [54]	Dengue	Peru
Fernandes-Matano et al., 2021 [55]	*Coronavirus 229E* *Virus sincitial respiratorio* *Influenza A* *Parainfluenza*	Mexico
Hazra et al., 2020 [56]	*Adenovirus* *Coronavirus NL63* *Human metapneumovirus* *Influenza A* *Parainfluenza*	USA
Hirotsu et al., 2020 [57]	*Rinovirus-enterovirus* *Metapneumovirus* *Coronavirus 229E* *Coronavirus OC43* *Adenovirus* *Virus sincitial respiratorio* *Coronavirus NL63*	Japan
Leuzinger et al., 2020 [58]	*Rinovirus* *Virus parainfluenza* *Influenza A* *Adenovirus* *Virus sincitial respiratorio*	Switzerland
Intra et al., 2020 [59]	*Candida albicans* *Candida glabrata* *Aspergillus fumigatus* *Staphylococcus aureus* *Streptococcus pyogenes* *Escherichia coli*	Italy
Nasir et al., 2020 [60]	*Klebsiella pneumoniae* *Pseudomonas aeruginosa* *Acinetobacter baumannii* *Staphylococcus aureus (MRSA)*	Pakistan
Lv et al., 2020 [61]	*Acinetobacter baumannii* *Escherichia coli* *Staphylococcus haemolyticus* *Pseudomonas aeruginosa* *Mycoplasma pneumoniae* *Stenotrophomonas maltophilia* *Enterococcus faecium* *Candida albicans* *Candida tropicalis* *Candida parapsilosis* *Candida lusitaniae*	China
Messina et al., 2020 [62]	*Histoplasma capsulatum*	Argentina

Molina et al. [63], who conducted a study in eight hospitals in Colombia, reported the most frequently seen microorganisms in coinfections in the ICU as follows: *Staphylococcus aureus*, *Streptococcus agalactiae*, and *Klebsiella pneumoniae* [63]. 

The most common coinfections with SARS-CoV2 in Colombia, as shown in Table 2, were dengue virus (DENV), *Klebsiella pneumoniae*, *Mycobacterium tuberculosis* (MTB), *Pneumocystis jirovecii* (*P. jirovecii*), *Cryptococcus neoformans*, *rhinovirus–enterovirus*, *adenovirus*, *human immunodeficiency virus* (HIV), and *Trypanosoma cruzi.*

## 7. SARS-CoV-2, and Dengue Coinfection

Initial differentiation between dengue and COVID-19 is a challenge as both infections have similar symptoms, such as fever, diarrhea, myalgia, and headache, and these overlap [64,65]. However, studies suggested that SARS-CoV-2 and DENV coinfection had less severe symptoms compared to isolated monoinfection [13]. Tropical countries, especially those with endemic DENV, approached a syndemic state, because multiple patients were coinfected with both SARS-CoV-2 and DENV. COVID-19 can be misdiagnosed with dengue. This situation complicates matters, so it can be difficult to distinguish between early infections and coinfections, generating an important risk for the population and demanding greater attention from health systems, because both viruses can cause serious complications, mainly through the cytokine storm in lung tissue caused by macrophage hyperactivation [13,27,64]. 

Diagnosing coinfection requires a combination of tests for the direct detection of the virus and indirect techniques that measure the immune response [65]. It is important to take into account the clinical and epidemiological particularities of both infections [13,65]. 

In 2020, 28,068 cases of dengue were reported in the department of Valle del Cauca, of which a case fatality rate of 6.6% was reported for severe dengue. It should be noted that from weeks 1 to 36, the cases were higher than expected, but from weeks 37 to 52, they were within the limit according to their historical behavior between 2013 and 2017 [63]. Likewise, in 2021, 8141 cases of dengue were reported, with 23 probable deaths due to severe dengue, although the number of cases was within the expected range [75]. However, a rebound peak of dengue was observed due to the replenishment of susceptible individuals with low exposure to infection. In addition, the number of patients requiring intensive care and mechanical ventilation increased. Therefore, regions such as Valle del Cauca should consider intensified preparedness for such scenarios, and further studies should be conducted to address this critical issue promptly to reduce the potential overload on the national health system [64].

## 8. Coinfections with Bacteria

*Klebsiella pneumoniae* is a Gram-negative species that can reside in the gastrointestinal tract [76]. In immunocompromised patients, it can cause serious infections, including urinary tract infections, respiratory infections, soft tissue infections, peritonitis, and sepsis [45,68]. Lipopolysaccharide and cell wall protein receptors are responsible for pathogenicity and determine the process of binding to host cells and provide protection against the human immune system response [76]. Mechanical ventilation, exposure to carbapenems and β-lactamase/β-lactamase inhibitors, renal replacement therapy, transfusions, and prolonged hospital stay are risk factors for coinfection with *Klebsiella pneumoniae* [77]. Patients with COVID-19, in whom immune mechanisms appear to be weakened by this viral infection, should follow rational antibiotic therapy with the aim of preventing bacterial resistance [76].

*Mycobacterium tuberculosis* (MTB) is an acid-fast bacterium and is the main causative agent of human tuberculosis. Evidence suggests a twofold increased mortality risk in patients with COVID-19 and tuberculosis [27]. For example, a systematic review reported an increased risk of mortality in patients with coinfection of these two microorganisms, although MTB and SARS-CoV-2 coinfections are poorly understood. On the other hand, a meta-analysis and systematic case study of SARS-CoV-2 coinfection with drug-resistant tuberculosis failed to show the same. In both cases, the evidence was inconsistent, and more high-quality studies are needed to better understand the causal association [51]. However, another study reported that active or latent tuberculosis increased susceptibility to *SARS-CoV-2* and disease severity [27]. 

## 9. Coinfections with Fungi

One of the underestimated microorganisms in patients with coronavirus disease 2019 is *Pneumocystis jirovecii*, an opportunistic infection that mainly affects immunosuppressed patients [52]. It occurs due to an imbalance between T lymphocyte subtypes, mainly due to the absence of CD4^+^ T cells, causing a deficiency in the immune response and generating a predominance of TCD8+, causing epithelial damage secondary to an excessive inflammatory state. In severe infections, 50% of patients may require hospitalization in the ICU, with a mortality rate of up to 40–60% [39,52]. SARS-CoV-2 infection can cause a state of immunodeficiency that may allow the appearance of this opportunistic fungus. At the same time, *P. jirovecii* and SARS-CoV-2 infections presented as joint processes, primarily in immunocompromised patients. It presents acutely with severe hypoxemia and the rapid deterioration of respiratory function, requiring invasive mechanical ventilation. It was found that the (1,3)-β-D-glucan detection technique is of potential use for the detection of *P. jirovecii* in patients with acute SARS-CoV-2 coinfection [39]. 

*Cryptococcus neoformans* is an encapsulated yeast-like fungus that is considered an opportunistic and rare pathogen in transplant recipients [50] and can cause cryptococcosis in immunocompromised patients. The most frequent and severe form of presentation is infection of the central nervous system, manifesting as subacute or chronic meningitis and characterized by headache, nausea, vomiting, fever, and altered consciousness; pulmonary, skin, lymph node, or other organ involvement may also occur to a lesser extent [66]. COVID-19 has been observed to increase infection by other rare pathogens in immunocompetent patients, such as that caused by *Aspergillus* spp., and studies have reported that patients required ICU stay and invasive mechanical ventilation secondary to infection by COVID-19, without a history of immunosuppression. Therefore, it can be said that coinfection with SARS-CoV-2 and Cryptococcus is rare. For example, in 2021, among 293 patients in a case study conducted in China, there was only one case reported [50,66]. However, it should be taken into account that SARS-CoV-2 infection may be an etiology for fungal infection by *Cryptococcus neoformans* due to the great multisystemic impact and the multiorgan dysfunction established by the viral agent, as the main cause of immunosuppression predisposing to infections by this microorganism. At the same time, management with glucocorticoids (dexamethasone) may favor immunological compromise [66].

## 10. Coinfection with Virus 

*Rhinoviruses* (*RVH*) and *enteroviruses* (*EVH*) belong to the *Picornaviridae* family and are the main cause of infections worldwide. They are characterized as small with a single-stranded RNA genome in an icosahedral capsid. In a study by Kim et al. [78], it was observed that 9.5% of 1217 patients with respiratory symptoms tested positive for SARS-CoV-2 and 318 for another microorganism. Of the group positive for SARS-CoV-2, 20.7% were positive for one or more pathogens, with *RVH* and EVH being the most frequent [78,79]. Regarding concomitant SARS-CoV-2 and *rhinovirus*–*enterovirus* infection, it is recommended that the multiplex PCR respiratory panel be performed only for severe patients and those in whom a positive result requires modification of treatment to prevent disease progression and even death [79]. 

*Adenoviruses* are icosahedral viruses that possess a double-stranded DNA genome, belonging to the *Adenoviridae* family. They cause only respiratory, ocular, and enteric disease in humans. Cases of coinfection with SARS-CoV-2 and respiratory viruses are poorly documented; for example, by 2020, only two patients with *adenovirus* coinfection had been documented. The pathogenesis of coinfection is not clearly explained, although there are hypotheses about the low presentation of SARS-CoV-2 without a predisposing risk factor. In addition, it has been documented that there is a higher proportion of coinfection in patients with acute respiratory distress syndrome and septic shock and requiring ICU admission [43]. 

HIV, which belongs to the *Retroviridae* family, is characterized by attacking the immune system and thus generating a state of immunodeficiency. People living with HIV (PLHIV) should not consider themselves protected against SARS-CoV-2, as people with low CD4^+^ T-cell counts may have worse outcomes than people with normal immunity [27,51]. Moreover, poorer COVID-related outcomes were observed in patients with HIV than those without the infection, especially in those with multimorbidity and advanced age [51]. A recent prospective cohort study reported that 8% of PLHIV infected with *SARS-CoV-2* required admission to the ICU. It should be noted that clinical manifestations, disease severity, and mortality are independent of HIV or antiretroviral-related factors. 

There are possible similarities between HIV*-1* and SARS-CoV-2 proteins. At the same time, it has not been identified as a common comorbidity in patients with COVID-19. In addition, there are differences between the receptors through which HIV and other pathogenic *coronaviruses* enter target cells, the ways of assembly, and their encapsulation. In the case of HIV, it does so near the cytoplasmic membrane, and in the case of *coronavirus*, the process takes place in the endoplasmic reticulum, which may suggest that there is no synergistic or cooperative pathogenesis [53]. 

## 11. Coinfections with Parasites

The protozoan parasite, *Trypanosoma cruzi*, is the main infectious agent of Chagas disease. This disease causes cardiac and gastrointestinal complications, among others, and is endemic in Latin America [80]. The intense inflammatory process of COVID-19 in immunocompromised patients could potentially influence the evolution of the disease and latently trigger reactivation of Chagas disease due to viral interference of the infection. It is important to highlight that different clinical and epidemiological scenarios may increase susceptibility to SARS-CoV-2 infection, because the new *coronavirus* disease has a significant impact on the heart. Likewise, the pandemic influences access to treatment for people with acute and chronic indeterminate Chagas disease [68]. 

In patients with SARS-CoV-2 pneumonia and immunosuppression, antibiotic treatment should be initiated in order to address opportunistic pulmonary infections. However, there are infections that are not covered by first-line antibiotics [39]. An example is *P. jirovecii*, where the first choice is the use of trimethoprim–sulfamethoxazole, a broad-spectrum antibiotic for bacterial and fungal germs [39,52]. Another case is *Klebsiella pneumoniae*, which has a very low sensitivity profile to most categories of antibiotics [14,38,45,76,77] and is very difficult to treat, but the drug combinations aztreonam and ceftazidime/avibactam or meropenem/vaborbactam show universal coverage against beta-lactamase-producing *Enterobacteriaceae*, including those with extensive drug resistance [76]. 

However, concerns about coinfections have led to significant antimicrobial use in up to 80% of critically ill patients with COVID-19, and their overuse could lead to antimicrobial resistance [38,63,81]. The rate of antimicrobial resistance ranged from 33.3% to 90.0%, depending on the infecting species [45]. The use of antibiotics over the last 50 years has exerted selective pressure on susceptible bacteria and may have favored the survival of resistant strains [15]. Of concern is that there is clinical evidence suggesting that inappropriate empirical use of antibiotics and other broad-spectrum antimicrobials may be associated with increased mortality [16]. One of the main aspects that needs to be evaluated in the prevalence of coinfections is the application of empirical antimicrobial treatment in patients with SARS-CoV-2 infection [38]. In addition, it is important to know more information about the prevalence of coinfections in the community [82], and hospital-acquired or healthcare-associated infections need to be continuously monitored and controlled. These should not only focus on minimizing the spread of SARS-CoV-2 infection but also on reducing bacterial cross-transmission, particularly of multidrug-resistant organisms [76]. Therefore, strategies should be established to improve antimicrobial stewardship in patients with COVID-19.

## 12. SARS-CoV-2—Associated Superinfections in Colombia

SARS-CoV-2 superinfections increase difficulties in the prognosis, diagnosis, and treatment of patients [83]. According to the U.S. Centers for Disease Control and Prevention, “a superinfection is an infection that follows a previous infection, especially when caused by pathogens that are resistant or have become resistant to previously used antibiotics” [6,84]. The mechanisms of superinfections include virus-induced damage to the respiratory system, decreased mucociliary clearance, and damage to the immune system. The decrease in lymphocytes and host immune function is the main reason that facilitates superinfection [83]. It should also be noted that clinical deterioration, elevated inflammatory markers, and bilateral radiological infiltrates may lead to misperception regarding the presence of a co-pathogen and should be used as an impetus to initiate comprehensive diagnostic workup with sampling, rather than as an indicator of underlying superinfection [84]. 

Superinfection in hospitalized patients with COVID-19 is associated with disease progression and poor prognosis. This situation increases antimicrobial treatment and mortality. They have even been related mainly to ICU admission, especially with the use of mechanical ventilation and catheters, and patients with comorbidities. In addition to having a higher prevalence, they also have a higher risk of death than in other patients [5,6,83]. The study by Clancy et al., (2021) [85] described several common risk factors, including being older than 60 years, male, ICU admission, mechanical ventilation, renal failure requiring hemodialysis, arterial hypertension, diabetes mellitus, and cancer [5,8,83]. 

Similarly, Paparoupa et al. [84] reported that 45% of invasively mechanically ventilated patients with COVID-19 pneumonia had bacterial, viral, or fungal respiratory superinfection in at least one of the sequential study periods. Therefore, the frequency of hospital-acquired superinfections remained low despite the fact that many patients received treatment that resulted in severe immunosuppression. 

Factors such as the empirical use of antibiotics, isolation measures, or the host macrophage activation explain this. At the same time, the lack of additional microbiological testing after SARS-CoV-2 was detected may have also contributed. Further studies will be needed to elucidate the role of each measure in reducing superinfections [5]. Table 3 lists the most frequent superinfections. *Acinetobacter* spp. has been identified as a common infection in ventilated patients. It is more frequent in patients with superinfection [6]. 

The superinfections with SARS-CoV-2 documented so far in Colombia, shown in Table 4, are bacterial in nature, with *Raoultella planticola* and *Pandoraea pnomenusa* standing out. 

*R. planticola* is a bacterium of the *Enterobacteriaceae* family that can be found in soil and water, and is associated with seafood consumption, biliary tract diseases, malignancy, diabetes mellitus, trauma, immunosuppression, and nosocomial infection. Initially considered harmless, the number of cases has increased in recent years, mainly consisting of cystitis, bacteremia, and pneumonia. Most strains of *R. planticola* are usually multisensitive and treatment is effective with second and third generation cephalosporins, aminoglycosides, and fluoroquinolones. With respect to *R. planticola* infection as a complication of SARS-CoV-2 infection, there is a paucity of cases reported in the literature. However, an infection by this microorganism has been reported as a complication of pulmonary bulla, a rare complication of COVID-19 affecting only 1% of patients, secondary to SARS-CoV-2 infection [94]. 

*Pandoraea pnomenusa* is a bacterium belonging to the *Pandoraea* genus. It should be noted that its usual presentation is the colonization of structurally abnormal airways. Infection generated by this microorganism occurs rarely, but its mortality rate is high, reaching up to 60%. It usually presents with multiple antimicrobial resistance. It presents an intrinsic ß-lactamase of the OXA type and a gyrB gene [95].

Bacterial superinfection in hospitalized patients with COVID-19 is associated with disease progression and poor prognosis. For example, in March 2020, a case of pneumonia caused by *Staphylococcus aureus* secreting leukocidin toxin in a man with mechanical ventilation was reported and treated with piperacillin–tazobactam, linezolid, meropenem, and gentamicin; however, the patient died 17 days after admission [8]. 

Antimicrobial stewardship will continue to be a priority because antimicrobial use in SARS-CoV-2 -infected patients remains higher than in superinfections [14]. Antimicrobial stewardship principles help guide the appropriate use of antibiotics [8]. Paparoupa et al. [84] demonstrated extensive use of broad-spectrum antibacterials in more than 70% of COVID-19 cases.

It is important that centers collect and publish their clinical, microbiological, and antimicrobial prescribing data. Further research is also needed on current infection control guidelines [8]. 

## 13. Therapeutic against COVID-19 in Colombia

In the context of COVID-19, antibiotics are primarily used to treat secondary bacterial infections that may occur as complications of the viral disease. These bacterial infections can manifest as secondary bacterial pneumonia, respiratory tract infections, or urinary tract infections, among others. The use of antibiotics in these cases aims to treat concurrent bacterial infections or prevent their occurrence in patients with compromised immune systems, particularly those who are in a severe condition [83]. In their study on antibiotic resistance during COVID-19 in Valle del Cauca, Colombia, Hurtado et al. [4] analyzed data from 31 hospitals and compared antibiotic resistance and consumption before (March 2018 to July 2019) and during (March 2020 to July 2021) the pandemic. The results showed an increase in the total number of bacterial isolates during the pandemic, accompanied by a significant decrease in resistance for four bug–drug combinations. However, there was a noticeable rise in vancomycin resistance among *Enterococcus faecium*. Overall, antibiotic consumption increased, except for meropenem in ICU settings. These findings suggest that the COVID-19 pandemic contributed to an increase in community-acquired infections, resulting in changes in antibiotic resistance patterns. Monitoring the increasing resistance of *E. faecium* to vancomycin and implementing effective infection control measures is crucial.

In a study conducted by Valladales-Restrepo et al., (2023) in Colombia, a descriptive cross-sectional study was carried out to examine the utilization of systemic antibiotics among patients diagnosed with COVID-19 between 2020 and 2022. The study involved eight clinics and included a total of 10,916 predominantly male patients with a median age of 57 years. Approximately 57.5% of the patients received antibiotics, with ampicillin/sulbactam and clarithromycin being the most frequently prescribed ones. Based on the WHO AWaRe classification, the majority of prescribed antibiotics belonged to the Watch category, followed by access and reserve categories. Several factors were found to be associated with a higher likelihood of receiving systemic antibiotics, such as male gender, older age, presence of dyspnea, rheumatoid arthritis, high blood pressure, in-hospital treatment, or ICU admission, and the use of systemic glucocorticoids, vasopressors, or invasive mechanical ventilation. Despite the low prevalence of bacterial coinfections, a significant proportion of COVID-19 patients received antibiotics, with a noticeable dominance of Watch antibiotics, which deviates from the recommendations provided by the World Health Organization [96].

In Colombia, various treatments have been used to combat COVID-19, including Remdesivir, Molnupiravir, Tocilizumab, and Convalescent Plasma Therapy. Remdesivir works by inhibiting the replication of the virus in the body and has been used in hospitalized patients with severe illness. Its emergency use authorization has made it available for those in need during the pandemic [97,98,99]. Molnupiravir is an antiviral medication that has shown activity against SARS-CoV-2 and has been authorized for emergency use in the treatment of COVID-19 by the Colombian regulatory agency, the National Institute for Surveillance of Drugs and Food (INVIMA) [100]. Preclinical studies have shown that Molnupiravir has the potential to significantly reduce viral load and decrease virus transmission in animal models. Additionally, early findings from clinical trials and systematic reviews have demonstrated promising results, indicating a reduction in hospitalization and mortality rates among patients with mild to moderate COVID-19. Molnupiravir presents a new ray of hope in combating the pandemic and could play a crucial role in the treatment of this disease [101,102,103].

Tocilizumab, on the other hand, is an immunosuppressive medication used in severe cases of COVID-19. This drug works by blocking interleukin-6 (IL-6), an inflammatory protein involved in the exaggerated immune response that can lead to severe complications in COVID-19 patients. Tocilizumab is administered via intravenous infusion and has been used in patients exhibiting excessive inflammatory response, such as cytokine release syndrome [104,105,106].

Convalescent Plasma Therapy is another treatment option used in Colombia for patients with COVID-19. It involves transfusing blood plasma from individuals who have recovered from the disease and have developed antibodies against the virus. It is believed that these antibodies can help fight the infection and improve symptoms in sick patients. However, it is important to note that the effectiveness of this therapy is still being investigated, and more scientific evidence is needed to support its widespread use [107,108].

## 14. Therapeutic Alternatives against COVID-19

The COVID-19 pandemic predisposes patients to potentially life-threatening infections in the ICU, hindering proper diagnosis and treatment [109]. Thus, specific therapy for COVID-19 should take into account coverage of local endemic pathogens that may occur in a similar manner, particularly while confirmation of SARS-CoV-2 infection is pending [17].

On the other hand, the WHO treatment guideline recommends empirically prescribed broad-spectrum antibiotics to treat possible coinfections. However, the effect of this respiratory disease on antimicrobial resistance is a dimension that requires necessary attention, since 15% to 50% of bacterial isolates are resistant to at least one antimicrobial group [109]. Therefore, antimicrobial therapy should be evaluated against a patient’s host factors and local epidemiology on a daily basis [17].

It has been documented that systemic glucocorticoids improve survival when administered to moderate or severe COVID patients. Its use is associated with reduced oxygen therapy and decreased risk of invasive mechanical ventilation among patients receiving supplemental oxygen [98,110]. However, treatment with corticosteroids was associated with a higher risk of progression of hospital stay and it is not clear if there is an increased risk of superinfection in non-severe COVID patients [110]. There are currently drugs or vaccines available to inhibit the new *coronavirus*. Although current vaccines effectively prevent serious complications and deaths, treatment options are still under validation, especially for immunocompromised patients [111]. Therefore, to avoid viral exposure, it is important to adhere to the following measures: maintaining social distancing, wearing face coverings, practicing frequent hand washing, using alcohol-based hand sanitizers when necessary, and avoiding touching the face. However, it is important to note that not all of these measures should be grouped together. While the effectiveness of alcohol-based hand sanitizers has been highlighted [17], it is also worth mentioning that the effectiveness of face coverings, such as masks, has been extensively studied. Additionally, it is crucial to be mindful of the potential contribution of these practices to the threat of antimicrobial resistance. Therefore, alternatives such as supplementation and the development of nanomedicine should be considered. 

Zinc (Zn) is involved in several biological processes as a cofactor and signaling molecule in the immune system. Additionally, it is an important component of the hormone thymulin, whose function is in T-cell differentiation, maturation, and NK actions [17]. Zn deficiency causes IL-10 dysregulation that alters Th1 cell response and macrophage functions [112]. Zn deficiency is associated with the risk and extreme progression of COVID-19 [17]. On the other hand, it has been shown that Zn ions can inhibit *coronavirus* RNA polymerase activity by reducing replication [112]. Thus, Zn supplementation may be associated with a lower mortality rate in patients with COVID-19, although more research is needed to understand the intimate mechanisms of antiviral activity [17].

Nanotechnology allows the manipulation and evaluation of individual molecules, and nanotechnology applied to medicine, also called nanomedicine, has been used to improve care in neurological, cardiovascular, and infectious diseases, and cancer [17]. Nanotechnology-based targets should be harnessed to aid in the fight against COVID-19, as well as any future pandemics, including the use of biosensors, virus inhibition by nanosystems, new vaccines and drugs, superfine filters for face masks or blood filters, and improvements to contact-tracing instruments, as it provides significant benefits and early-stage disease detection. In addition, established methods do not require specialized instrumentation, which provides a pathway to simple integral answers [113,114]. 

Another alternative is the use of REGEN-COV for patients who are at high risk of progressing to severe COVID-19 and who are not fully vaccinated or are not expected to develop a complete immune response [98]. This combination of monoclonal antibodies has been shown to reduce the viral load in symptomatic outpatient patients and the number of medical visits. It also has in vitro activity against current strains and reduces the risk of hospitalization or death. However, it may be associated with worse outcomes when given to patients who require high-flow oxygen or mechanical ventilation [98,115]. 

## 15. Conclusions

The COVID-19 pandemic has had a significant impact on Latin America, particularly on countries with weaker healthcare systems. The disease has disproportionately affected the elderly and disadvantaged, and has been exacerbated by comorbidities such as hypertension, diabetes, and respiratory diseases. Coinfections with other microorganisms, such as bacteria, viruses, and fungi, have also contributed to the morbidity and mortality rates of COVID-19 in the region. In Colombia, dengue virus, *Klebsiella pneumoniae*, *Mycobacterium tuberculosis*, *Pneumocystis jirovecii*, *Cryptococcus neoformans*, and *rhinovirus–enterovirus* have been reported as common coinfections with SARS-CoV-2. There is a need to improve surveillance systems and establish protocols for the early detection and management of coinfections in COVID-19 to reduce the burden on healthcare systems. While current vaccines have been effective in preventing serious complications and deaths, there is a lack of specific treatment options for COVID-19, especially for immunocompromised patients. Alternative approaches, such as zinc supplementation and the use of nanomedicine, have been proposed as potential therapies, but more research is needed to fully understand their effectiveness.

## Figures and Tables

**Figure 1 medicina-59-01336-f001:**
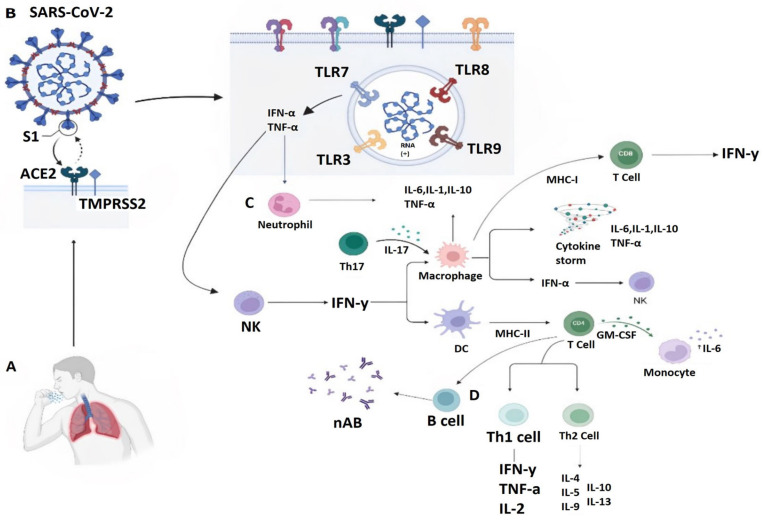
(**A**) Primary mode of transmission of SARS-CoV-2, which is through respiratory droplets. (**B**) SARS-CoV-2 enters via the angiotensin-converting enzyme receptor 2 (ACE-2). It also shows toll-like receptors whose function is to recognize viral RNA in endosomes. (**C**) Activation of the antiviral innate immune response associated with IFN production and activation of proinflammatory cytokines mediated by T lymphocytes causing a cytokine storm. (**D**) Humoral response to SARS-CoV-2 with the use of neutralizing antibodies (nAb). The figure was created with https://app.biorender.com (accessed on 1 June 2023).

**Table 2 medicina-59-01336-t002:** Coinfections with *SARS-CoV-2* in Colombia.

Author	Coinfection	City/State
Algarín-Lara et al., 2021 [39]	*Pneumocystis jirovecii*	Barranquilla
Motta et al., 2022 [43]	*Adenovirus*	Bogota
Torres-Serrano et al., 2021 [50]	*Cryptococcus neoformans*	Bogota
Ortiz-Martínez et al., 2021 [51]	*Mycobacterium tuberculosis*	Bucaramanga
Angel et al., 2020 [53]	VIH	Santiago de Cali
Cardona-Ospina et al., 2021 [64]	Dengue	Valle del Cauca
Agudelo Rojas et al., 2020 [65]	Dengue	Cali
Restrepo et al., 2021 [66]	*Cryptococcus neoformans*	Bogota
León et al., 2022 [67]	*Trypanosoma cruzi*	Bucaramanga
Villamil-Gómez et al., 2021 [68]	*Trypanosoma cruzi*	Sincelejo
Medina-Ahumada et al., 2022 [69]	*Aspergillus fumigatus*	Cartagena
Molina et al., 2022 [63]	*Enterobacter cloacae* *Haemophilus influenzae* *Klebsiella pneumoniae* *Klebsiella oxytoca* *Pseudomonas aeruginosa* *Streptococcus agalactiae* *Staphylococcus aureus* *Streptococcus pneumoniae*	Medellin
Pinzón et al., 2021 [70]	*Mycoplasma pneumoniae* *Salmonella* spp. *Mycobacterium tuberculosis* *Pneumococcus* *Influenza* *Bordetella* spp.	Antioquia
Molano et al., 2021 [71]	*Klebsiella pneumoniae*	Bogota
García-Posada et al., 2021 [72]	*Klebsiella pneumoniae* *Pseudomonas aeruginosa* *Staphylococcus aureus* *Enterobacter cloacae* *Rhinovirus-enterovirus*	Cordoba
Orozco-Hernandez et al., 2020 [73]	*Rinovirus-enterovirus*	Cartago
Contreras-Acosta et al., 2020 [74]	*Influenza H1N1*	Barranquilla

**Table 3 medicina-59-01336-t003:** Superinfections with SARS-CoV-2.

Author	Superinfection	Country
García-Vidal et al., 2021 [5]	*Staphylococcus aureus* *Stenotrophomonas maltophilia* *Pseudomonas aeruginosa* *Klebsiella pneumoniae*	Spain
Musuuza et al., 2021 [6]	*Acinetobacter* spp. *Pseudomonas* spp. *Escherichia coli* *Rhinovirus* *Candida* spp.	USA
Nag et al., 2021 [8]	*Klebsiella pneumoniae* *Pseudomonas aeruginosa* *Serratia marcescens* *Enterobacter cloacae* *Acinetobacter baumannii* *Escherichia coli* *Staphylococcus aureus* *Bacillus cereus* *Aspergillus flavus* *Aspergillus fumigatus* *Candida albicans* *Candida glabrata*	India
Al-Tawfiq et al., 2021 [86]	*Mucorales*	
Arcangeletti et al., 2022 [87]	*Influenza A H3*	Italy
Vaseghi et al., 2022 [88]	*Candida auris*	Iran
Wertheim et al., 2022 [89]	Alpha and Epsilon del SARS-CoV-2	USA
Pickens et al., 2021 [90]	*Methicillin-susceptible* *Staphylococcus aureus (MSSA)* *Streptococcus agalactiae* *Stenotrophomonas maltophilia* *Methicillin-resistant* *Staphylococcus aureus (MRSA)* *Pneumocystis* spp. *Haemophilus influenzae*	
Awada et al., 2021 [91]	*Candida duobushaemulonii*	Lebanon
Lamballerie et al., 2020 [92]	*Aspergillus* spp.	France
Nieuwenhuis MB et al., 2020 [93]	*Staphylococcus aureus* *Pseudomonas aeruginosa*	Netherlands

**Table 4 medicina-59-01336-t004:** Superinfections with SARS-CoV-2 in Colombia.

Author	Superinfection	City/State
Castaño-Correa et al., 2021 [83]	*Klebsiella pneumoniae* *Staphylococcus aureus* *Enterobacter cloacae* *Enterobacter aerogenes* *Pseudomonas aeruginosa* *Serratia marcescens* *Haemophilus influenzae* *Escherichia coli*	Medellin
Montalvo et al., 2022 [94]	*Raoultella planticola*	Bogota
Cubides-Diaz et al., 2022 [95]	*Pandoraea pnomenusa*	Cundinamarca

## Data Availability

Not applicable.

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
