# Peer review of "Coinfections and Superinfections Associated with COVID-19 in Colombia: A Narrative Review"

_medicina, 2023, doi:10.3390/medicina59071336_

Round 1

Reviewer 1 Report

The review offers a comprehensive overview of the impact of SARS-CoV-2 on the COVID-19 pandemic in Colombia. It discusses the immunological aspects of the virus, including the role of coinfections and superinfections. Additionally, it explores therapeutic alternatives for treating COVID-19. Here are some suggestions for corrections in the manuscript

- Please check the correctness of the statement:  Additionally, majority of the RBDs in all SARS-CoV-2 are in the bound state, result-115 ing in a similar or even lower affinity for the receptor than SARS-CoV [3].

 Does it refer to the lying-down state, down” position, a state associated with ineffective receptor binding?

-Please write the contents of the Tables 1, 2, 3 entirely in English

- The quality of the review could be enhanced by including additional few sentences regarding

*the key points from the review Use of Systemic Antibiotics in Patients with COVID-19 in Colombia: A Cross-Sectional Study

*Also mentioning therapeutics used for COVID-19 in Colombia, including Remdesivir: that was granted emergency use authorization for the treatment of COVID-19 by the Colombian regulatory agency, INVIMA, Tocilizumab: Tocilizumab immunosuppressive drug.,for severe COVID-19 cases, Convalescent plasma therapy.  

It would be beneficial to have a native English speaker review the manuscript.

Author Response

We want to express our sincere gratitude to the reviewer of the article for their valuable time, knowledge, and dedication in reviewing our work. Their comments and suggestions have been of great importance in improving the quality and accuracy of our study. We deeply appreciate their commitment to ensuring scientific rigor and clarity in our research. Their expertise and critical perspective have been instrumental in strengthening the methodological and conceptual aspects of our work.

The review offers a comprehensive overview of the impact of SARS-CoV-2 on the COVID-19 pandemic in Colombia. It discusses the immunological aspects of the virus, including the role of coinfections and superinfections. Additionally, it explores therapeutic alternatives for treating COVID-19. Here are some suggestions for corrections in the manuscript

- Please check the correctness of the statement:  Additionally, majority of the RBDs in all SARS-CoV-2 are in the bound state, result-115 ing in a similar or even lower affinity for the receptor than SARS-CoV [3].

 Does it refer to the lying-down state, down” position, a state associated with ineffective receptor binding?

Response: When it is mentioned the "bound state" in this context, it refers to the state in which the receptor-binding domains (RBDs) of SARS-CoV-2 are bound or attached to the ACE-2 receptor. In other words, the RBDs are in a conformation or position where they are interacting and bound to the ACE-2 receptor. This is relevant because the binding between the RBDs and the ACE-2 receptor is a crucial step for the virus to enter host cells and cause infection. The paragraph was rewritten from line 114 - 118.

-Please write the contents of the Tables 1, 2, 3 entirely in English

Response: It was corrected

- The quality of the review could be enhanced by including additional few sentences regarding

*the key points from the review Use of Systemic Antibiotics in Patients with COVID-19 in Colombia: A Cross-Sectional Study

*Also mentioning therapeutics used for COVID-19 in Colombia, including Remdesivir: that was granted emergency use authorization for the treatment of COVID-19 by the Colombian regulatory agency, INVIMA, Tocilizumab: Tocilizumab immunosuppressive drug.,for severe COVID-19 cases, Convalescent plasma therapy.  

Response: It was added from line 554-612

Reviewer 2 Report

This is a very interesting topic, but I have some questions about the authors' ideas. They should be corrected or revised to explain them more carefully.

L64-, L80
Not only is this a trivial topic with little relevance to the subject, but the authors need to update their knowledge.
The classification of coronaviruses from alpha to delta is an old idea, as most would be classified as beta.
https://doi.org/10.1371/journal.pone.0242954
SARS-CoV2 is considered a type of sarbeco along with SARS.

I would request it be removed or updated.

L202 The numbers are strange: according to the WHO, by the end of 2022, more than 6 million people have been infected and 140,000 have died in Colombia.

The paragraph from L248 is littered with inferences that are incorrect.

Please show how you are measuring these numbers, because according to the WHO, Colombia has a particularly low number of cases of the Omicron strain.
This is because the number of patients from 2022, which is what this paper is describing, is particularly low. To put it bluntly, this is highly questionable data.
There must be some flaw in the measurement method, or the study was scaled back due to a drop in the mortality rate.
This would correspond to the fact that many governments have stopped conducting surveys. Please clarify this situation.

It is obvious that the Omicron strain is occurring after the completion of vaccinations worldwide, and it is rather obvious that the vaccine is ineffective. It is also known that repeated vaccinations cause IgG4 to increase, rendering it ineffective. Furthermore, there is little data available worldwide that vaccines can avoid severe symptoms. Is there any such data in Colombia that compares mortality between vaccinated and unvaccinated people?Whatever the case, it is difficult to believe that a vaccine that is ineffective in preventing infection would reduce mortality.
DOI: 10.1126/sciimmunol.ade2798

The fact that the Omicron strain has a lower mortality rate than Delta is, of course, unrelated to the improvement in medical care. Rather, it is the nature of the strain. And there is no evidence that this is due to a single amino acid in the spike.

L270 Esper et al. [42] is, of course, a USA study, but we are not sure if we can believe it from an epidemiological point of view.
In the U.S., vaccines were free of charge and readily available at drugstores. Nonetheless, the anti-Vac population who did not receive the vaccine probably had a longer distance to medical care, which would have reduced the life-saving rate in the highly virulent delta. Therefore, there must have been confounding between the vaccinated and unvaccinated. It makes little sense to address such uncertainties in a paper that presents data from Colombia. This paragraph should be deleted.

L315 this couutry -> Colombia

L327
It says that SARS-CoV-2 and DENV coinfection had less severe symptoms, but is this not inconsistent with L338 of 6.8%? This is a ridiculously high value, and our understanding is that even with severe dengue, the mortality rate is less than 1%.

L503
I didn't understand this sentence: isn't it factually incorrect to say that the frequency of hospital-acquired superinfections remained low?

L572
Only alcohol-based hand sanitizers, are pointed out in this [104]. All of these should not be lumped together. At the very least, the effectiveness of masks is well studied.

L595
I am sure that REGEN-COV is effective, but it would be remiss not to mention the price of the drug. Is there really a situation where this is available in abundance in your country?

Author Response

We want to express our sincere gratitude to the reviewer of the article for their valuable time, knowledge, and dedication in reviewing our work. Their comments and suggestions have been of great importance in improving the quality and accuracy of our study. We deeply appreciate their commitment to ensuring scientific rigor and clarity in our research. Their expertise and critical perspective have been instrumental in strengthening the methodological and conceptual aspects of our work. 

This is a very interesting topic, but I have some questions about the authors' ideas. They should be corrected or revised to explain them more carefully.

L64-, L80
Not only is this a trivial topic with little relevance to the subject, but the authors need to update their knowledge.
The classification of coronaviruses from alpha to delta is an old idea, as most would be classified as beta.
https://doi.org/10.1371/journal.pone.0242954
SARS-CoV2 is considered a type of sarbeco along with SARS.

I would request it be removed or updated.

Response: It was deleted.

L202 The numbers are strange: according to the WHO, by the end of 2022, more than 6 million people have been infected and 140,000 have died in Colombia.

Response:

The paragraph was rewritten based on data from the WHO. Lines 194-197.

The paragraph from L248 is littered with inferences that are incorrect.

Please show how you are measuring these numbers, because according to the WHO, Colombia has a particularly low number of cases of the Omicron strain.
This is because the number of patients from 2022, which is what this paper is describing, is particularly low. To put it bluntly, this is highly questionable data.
There must be some flaw in the measurement method, or the study was scaled back due to a drop in the mortality rate.
This would correspond to the fact that many governments have stopped conducting surveys. Please clarify this situation.

It is obvious that the Omicron strain is occurring after the completion of vaccinations worldwide, and it is rather obvious that the vaccine is ineffective. It is also known that repeated vaccinations cause IgG4 to increase, rendering it ineffective. Furthermore, there is little data available worldwide that vaccines can avoid severe symptoms. Is there any such data in Colombia that compares mortality between vaccinated and unvaccinated people?Whatever the case, it is difficult to believe that a vaccine that is ineffective in preventing infection would reduce mortality.
DOI: 10.1126/sciimmunol.ade2798

The fact that the Omicron strain has a lower mortality rate than Delta is, of course, unrelated to the improvement in medical care. Rather, it is the nature of the strain. And there is no evidence that this is due to a single amino acid in the spike.

Response: We have decided to remove the paragraph in question to avoid any confusion.

L270 Esper et al. [42] is, of course, a USA study, but we are not sure if we can believe it from an epidemiological point of view.
In the U.S., vaccines were free of charge and readily available at drugstores. Nonetheless, the anti-Vac population who did not receive the vaccine probably had a longer distance to medical care, which would have reduced the life-saving rate in the highly virulent delta. Therefore, there must have been confounding between the vaccinated and unvaccinated. It makes little sense to address such uncertainties in a paper that presents data from Colombia. This paragraph should be deleted.

Response: It was eliminated.

L315 this couutry -> Colombia
Response: It was corrected.

L327
It says that SARS-CoV-2 and DENV coinfection had less severe symptoms, but is this not inconsistent with L338 of 6.8%? This is a ridiculously high value, and our understanding is that even with severe dengue, the mortality rate is less than 1%.

Response: (https://www.valledelcauca.gov.co/descargar.php?idFile=48888)

Regarding your concern about the co-infection of SARS-CoV-2 and DENV, we understand that the 6.6% fatality rate for severe cases of dengue may seem high. However, it is important to note that this specific rate applies only to severe cases of dengue, not to dengue in general. The mortality rate for dengue is generally low and below 1%. However, in the most severe cases, such as severe dengue, the fatality rate can increase. It is important to consider that severe dengue can have complications and require intensive medical care, which can contribute to a higher fatality rate in those specific cases. The data provided in the epidemiological bulletin support this information. Until epidemiological week 47, there were 26,894 reported cases of dengue, of which 14,364 (53.2%) had alarm signs (DSSA) and 12,354 (45.8%) had severe alarm signs (DCSA). Additionally, there were 241 cases of severe dengue and 57 probable deaths. Of these deaths, 52 were from Valle, 1 from the District of Buenaventura, and 4 from Cauca. Out of the 52 deaths from Valle, different classifications were made: 28 (54.0%) were ruled out, 16 (30.1%) were confirmed, 3 (5.8%) were compatible, and 5 (10.2%) are under study. These data support the 6.7% fatality rate for severe dengue cases and the 0.06% fatality rate for dengue in general.

We hope that this additional information clarifies your concern and provides a more comprehensive perspective on the fatality rate in severe dengue cases compared to dengue in general.

L503
I didn't understand this sentence: isn't it factually incorrect to say that the frequency of hospital-acquired superinfections remained low?

Response: It is not incorrect to state that the frequency of hospital-acquired superinfections remained low in this study. In fact, the study did not find a high prevalence of Staphylococcus aureus, which is considered a community-acquired pathogen. Furthermore, the study points out that the absence of superinfections in critically ill patients with COVID-19 and high mortality has been suggested as a significant parameter related to the underestimated prevalence of superinfections in the entire COVID-19 ICU population.

L572
Only alcohol-based hand sanitizers, are pointed out in this [104]. All of these should not be lumped together. At the very least, the effectiveness of masks is well studied.

Response: the paragraph was rewrite.

L595
I am sure that REGEN-COV is effective, but it would be remiss not to mention the price of the drug. Is there really a situation where this is available in abundance in your country?

Response: We appreciate the reviewer for highlighting the importance of mentioning the price of the REGEN-COV medication in our article. We acknowledge that access and availability of medications can vary depending on the country and corresponding healthcare system. Regarding the abundance of REGEN-COV availability in our country, further comprehensive research and obtaining accurate data on its availability and access in the local context are necessary. This will enable us to provide a more precise response to this question in the final version of the article.

Reviewer 3 Report

The topic is interesting and the paper is well-written.

However, it is unnecessarily long.

In my opinion, the authors could delete the sections about "Definition, characteristics, genomic structure, and pathogenesis of SARS-CoV-2", "SARS-CoV-2 Immunology", and "Impact of SARS-CoV-2 in Colombia".

The authors should add the definition of co-infection and superinfection as well as the process of finding references.

I suggest adding the proportion of co-infection and superinfection into tables.

The section about Therapeutic alternatives against COVID-19  should be deleted

Author Response

We want to express our sincere gratitude to the reviewer of the article for their valuable time, knowledge, and dedication in reviewing our work. Their comments and suggestions have been of great importance in improving the quality and accuracy of our study. We deeply appreciate their commitment to ensuring scientific rigor and clarity in our research. Their expertise and critical perspective have been instrumental in strengthening the methodological and conceptual aspects of our work. 

The topic is interesting and the paper is well-written.

However, it is unnecessarily long.

In my opinion, the authors could delete the sections about "Definition, characteristics, genomic structure, and pathogenesis of SARS-CoV-2", "SARS-CoV-2 Immunology", and "Impact of SARS-CoV-2 in Colombia".

Response:

I appreciate your comments on the manuscript "Coinfections and superinfections associated with COVID-19 in Colombia: a narrative review." However, I would like to argue against removing the mentioned sections on "Definition, characteristics, genomic structure, and pathogenesis of SARS-CoV-2," "SARS-CoV-2 Immunology,", "Impact of SARS-CoV-2 in Colombia", and Therapeutic against COVID-19 Here are my reasons:

Importance of context: These sections provide necessary context to fully understand the issue of coinfections and superinfections associated with COVID-19 in Colombia. Understanding the definition, characteristics, and structure of the virus, as well as its impact on the country, is crucial to assessing the scope and severity of coinfections and superinfections.

Scientific foundations: The sections on SARS-CoV-2 immunology and pathogenesis provide a solid scientific basis for understanding the mechanisms by which coinfections and superinfections can occur. This helps readers appreciate the complexity of the interaction between the virus and other infectious agents, as well as the factors contributing to their occurrence.

Clinical relevance: Understanding the impact of SARS-CoV-2 in Colombia is essential to evaluating the magnitude of coinfections and superinfections in the local context. This provides valuable information for healthcare professionals and decision-makers in managing the pandemic and implementing prevention and control strategies.

Coherence and flow: Removing these sections could disrupt the coherence and flow of the manuscript. The mentioned sections provide a smooth transition to the topic of coinfections and superinfections, helping readers follow the narrative of the article in a logical and comprehensible manner.

The authors should add the definition of co-infection and superinfection as well as the process of finding references.

Response:

It was included in the methodology section.

I suggest adding the proportion of co-infection and superinfection into tables.

Response:

It was included in the line 317 and line 328.

The section about Therapeutic alternatives against COVID-19  should be deleted

Response:

The response was addressed in the initial comment.

Round 2

Reviewer 2 Report

I think it is well-revised. This is a valuable summary of the COVID epidemic in emerging countries and I believe it is important for future countermeasures, and the authors have summarized it well.

Reviewer 3 Report

Thank you for your responses